# The Return of the Gating Network: Combining Generative Models and Discriminative Training in Natural Image Priors

**Dan Rosenbaum**
School of Computer Science and Engineering
Hebrew University of Jerusalem

**Yair Weiss**
School of Computer Science and Engineering
Hebrew University of Jerusalem

## Abstract

In recent years, approaches based on machine learning have achieved state-of-the-art performance on image restoration problems. Successful approaches include both generative models of natural images as well as discriminative training of deep neural networks. Discriminative training of feed forward architectures allows explicit control over the computational cost of performing restoration and therefore often leads to better performance at the same cost at run time. In contrast, generative models have the advantage that they can be trained once and then adapted to any image restoration task by a simple use of Bayes' rule.

In this paper we show how to combine the strengths of both approaches by training a discriminative, feed-forward architecture to predict the state of latent variables in a generative model of natural images. We apply this idea to the very successful Gaussian Mixture Model (GMM) of natural images. We show that it is possible to achieve comparable performance as the original GMM but with two orders of magnitude improvement in run time while maintaining the advantage of generative models.

## 1 Introduction

Figure 1 shows an example of an image restoration problem. We are given a degraded image (in this case degraded with Gaussian noise) and seek to estimate the clean image. Image restoration is an extremely well studied problem and successful systems for specific scenarios have been built without any explicit use of machine learning. For example, approaches based on "coring" can be used to successfully remove noise from an image by transforming to a wavelet basis and zeroing out coefficients that are close to zero [7]. More recently the very successful BM3D method removes noise from patches by finding similar patches in the noisy image and combining all similar patches in a nonlinear way [4].

In recent years, machine learning based approaches are starting to outperform the hand engineered systems for image restoration. As in other areas of machine learning, these approaches can be divided into *generative* approaches which seek to learn probabilistic models of clean images versus *discriminative* approaches which seek to learn models that map noisy images to clean images while minimizing the training loss between the predicted clean image and the true one.

Two influential generative approaches are the fields of experts (FOE) approach [16] and KSVD [5] which assume that filter responses to natural images should be sparse and learn a set of filters under this assumption. While very good performance can be obtained using these methods, when they are trained generatively they do not give performance that is as good as BM3D. Perhaps the most successful generative approach to image restoration is based on Gaussian Mixture Models (GMMs) [22]. In this approach 8x8 image patches are modeled as 64 dimensional vectors and a

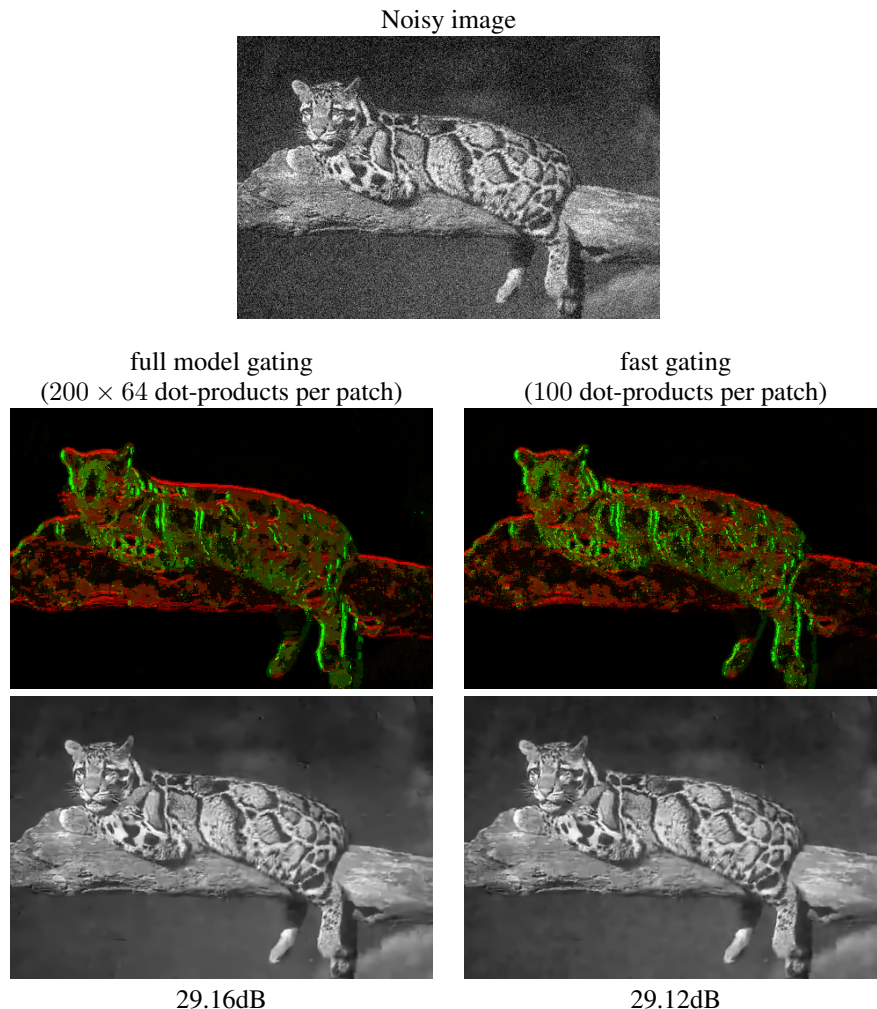

Noisy image

full model gating
($200 \times 64$ dot-products per patch)

fast gating
(100 dot-products per patch)

29.16dB

29.12dB

Figure 1: Image restoration with a Gaussian mixture model. Middle: the most probable component of every patch calculated using a full posterior calculation vs. a fast gating network (color coded by embedding in a 2-dimensional space). Bottom: the restored image: the gating network achieves almost identical results but in 2 orders of magnitude faster.

simple GMM with 200 components is used to model the density in this space. Despite its simplicity, this model remains among the top performing models in terms of likelihood given to left out patches and also gives excellent performance in image restoration [23, 20]. In particular, it outperforms BM3D on image denoising and has been successfully used for other image restoration problems such as deblurring [19]. The performance of generative models in denoising can be much improved by using an "empirical Bayes" approach where the parameters are estimated from the noisy image [13, 21, 14, 5].

Discriminative approaches for image restoration typically assume a particular feed forward structure and use training to optimize the parameters of the structure. Hel-Or and Shaked used discriminative training to optimize the parameters of coring [7]. Chen et al. [3] discriminatively learn the parameters of a generative model to minimize its denoising error. They show that even though the model was trained for a specific noise level, it acheives similar results as the GMM for different noise levels. Jain and Seung trained a convolutional deep neural network to perform image denoising. Using the same training set as was used by the FOE and GMM papers, they obtained better results than FOE but not as good as BM3D or GMM [9]. Burger et al. [2] trained a deep (non-convolutional) multi layer perceptron to perform denoising. By increasing the size of the training set by two orders of magnitude relative to previous approaches, they obtained what is perhaps the

best stand-alone method for image denoising. Fanello et al. [6] trained a random forest architecture to optimize denoising performance. They obtained results similar to the GMM but at a much smaller computational cost.

Which approach is better, discriminative or generative? First it should be said that the best performing methods in both categories give excellent performance. Indeed, even the BM3D approach (which can be outperformed by both types of methods) has been said to be close to optimal for image denoising [12]. The primary advantage of the discriminative approach is its *efficiency* at run-time. By defining a particular feed-forward architecture we are effectively constraining the computational cost at run-time and during learning we seek the best performing parameters for a fixed computational cost. The primary advantage of the generative approach, on the other hand, is its *modularity*. Learning only requires access to clean images, and after learning a density model for clean images, Bayes' rule can be used to peform restoration on any image degradation and can support different loss functions at test time. In contrast, discriminative training requires separate training (and usually separate architectures) for every possible image degradation. Given that there are literally an infinite number of ways to degrade images (not just Gaussian noise with different noise levels but also compression artifacts, blur etc.), one would like to have a method that maintains the modularity of generative models but with the computational cost of discriminative models.

In this paper we propose such an approach. Our method is based on the observation that the most costly part of inference with many generative models for natural images is in estimating latent variables. These latent variables can be abstract representations of local image covariance (e.g. [10]) or simply a discrete variable that indicates which Gaussian most likely generated the data in a GMM. We therefore discriminatively train a feed-forward architecture, or a "gating network" to predict these latent variables using far less computation. The gating network need only be trained on "clean" images and we show how to combine it during inference with Bayes' rule to perform image restoration for any type of image degradation. Our results show that we can maintain the accuracy and the modularity of generative models but with a speedup of two orders of magnitude in run time.

In the rest of the paper we focus on the Gaussian mixture model although this approach can be used for other generative models with latent variables like the one proposed by Karklin and Lewicki [10]. Code implementing our proposed algorithms for the GMM prior and Karklin and Lewicki's prior is available online at `www.cs.huji.ac.il/~danrsm`.

## 2 Image restoration with Gaussian mixture priors

Modeling image patches with Gaussian mixtures has proven to be very effective for image restoration [22]. In this model, the prior probability of an image patch $x$ is modeled by: $\Pr(x) = \sum_h \pi_h \mathcal{N}(x; \mu_h, \Sigma_h)$. During image restoration, this prior is combined with a likelihood function $\Pr(y|x)$ and restoration is based on the posterior probability $\Pr(x|y)$ which is computed using Bayes' rule. Typically, MAP estimators are used [22] although for some problems the more expensive BLS estimator has been shown to give an advantage [17].

In order to maximize the posterior probability different numerical optimizations can be used. Typically they require computing the *assignment probabilities*:

$$\Pr(h|x) = \frac{\pi_h \mathcal{N}(x; \mu_h, \Sigma_h)}{\sum_k \pi_k \mathcal{N}(x; \mu_k, \Sigma_k)} \tag{1}$$

These assignment probabilities play a central role in optimizing the posterior. For example, it is easy to see that the gradient of the log of the posterior involves a weighted sum of gradients where the assignment probabilities give the weights:

$$\begin{aligned}\frac{\partial \log \Pr(x|y)}{\partial x} &= \frac{\partial \left[\log \Pr(x) + \log \Pr(y|x) - \log \Pr(y)\right]}{\partial x} \\ &= -\sum_h \Pr(h|x)(x - \mu_h)^\top \Sigma_h^{-1} + \frac{\partial \log \Pr(y|x)}{\partial x}\end{aligned} \tag{2}$$

Similarly, one can use a version of the EM algorithm to iteratively maximize the posterior probability by solving a sequence of reweighted least squares problems. Here the assignment probabilities define the weights for the least squares problems [11]. Finally, in auxiliary samplers for performing

BLS estimation, each iteration requires sampling the hidden variables according to the current guess of the image [17].

For reasons of computational efficiency, the assignment probabilities are often used to calculate a hard assignment of a patch to a component:

$$\hat{h}(x) = \arg \max_h \Pr(h|x) \tag{3}$$

Following the literature on "mixtures of experts" [8] we call this process *gating*. As we now show, this process is often the most expensive part of performing image restoration with a GMM prior.

## 2.1 Running time of inference

The successful EPLL algorithm [22] for image restoration with patch priors defines a cost function based on the simplifying assumption that the patches of an image are independent:

$$J(x) = -\sum_i \log \Pr(x_i) - \lambda \log \Pr(y|x) \tag{4}$$

where $\{x_i\}$ are the image patches, $x$ is the full image and $\lambda$ is a parameter that compensates for the simplifying assumption. Minimizing this cost when the prior is a GMM, is done by alternating between three steps. We give here only a short representation of each step but the full algorithm is given in the supplementary material. The three steps are:

- Gating. For each patch, the current guess $x_i$ is assigned to one of the components $\hat{h}(x_i)$
- Filtering. For each patch, depending on the assignments $\hat{h}(x_i)$, a least squares problem is solved.
- Mixing. Overlapping patches are averaged together with the noisy image $y$.

It can be shown that after each iteration of the three steps, the EPLL splitting cost function (a relaxation of equation 4) is decreased.

In terms of computation time, the gating step is by far the most expensive one. The filtering step multiplies each $d$ dimensional patch by a single $d \times d$ matrix which is equivalent to $d$ dot-products or $d^2$ flops per patch. Assuming a local noise model, the mixing step involves summing up all patches back to the image and solving a local cost on the image (equivalent to 1 dot-product or $d$ flops per patch).[1] In the gating step however, we compute the probability of all the Gaussian components for every patch. Each computation performs $d$ dot-products, and so for $K$ components we get a total of $d \times K$ dot-products or $d^2 \times K$ flops per patch. For a GMM with 200 components like the one used in [22], this results in a gating step which is 200 times slower than the filtering and mixing steps.

## 3 The gating network

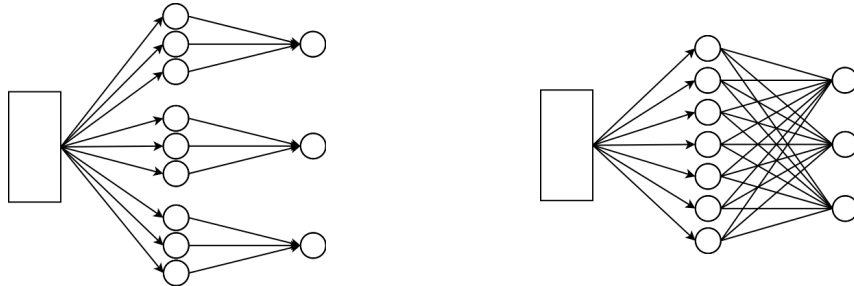

Figure 2: Architecture of the gating step in GMM inference (left) vs. a more efficient gating network.

The left side of figure 2 shows the computation involved in a naive computing of the gating. In the GMM used in [22], the Gaussians are zero mean so computing the most likely component involves multiplying each patch with all the eigenvectors of the covariance matrix and squaring the results:

$$\log Pr(x|h) = -x^{\top}\Sigma_h^{-1}x + const_h = -\sum_i \frac{1}{\sigma_i^h}(v_i^h x)^2 + const_h \tag{5}$$

where $\sigma_i^h$ and $v_i^h$ are the eigenvalues and eigenvectors of the covariance matrix. The eigenvectors can be viewed as templates, and therefore, the gating is performed according to weighted sums of dot-products with different templates. Every component has a different set of templates and a different weighting of their importance (the eigenvalues). Framing this process as a feed-forward network starting with a patch of dimension $d$ and using $K$ Gaussian components, the first layer computes $d \times K$ dot-products (followed by squaring), and the second layer performs $K$ dot-products.

Viewed this way, it is clear that the naive computation of the gating is inefficient. There is no "sharing" of dot-products between different components and the number of dot-products that are required for deciding about the appropriate component, may be much smaller than is done with this naive computation.

## 3.1 Discriminative training of the gating network

In order to obtain a more efficient gating network we use discriminative training. We rewrite equation 5 as:

$$\log Pr(x|h) \approx -\sum_i w_i^h (v_i^T x)^2 + const_h \tag{6}$$

Note that the vectors $v_i$ are required to be shared and do not depend on $h$. Only the weights $w_i^h$ depend on $h$.

Given a set of vectors $v_i$ and the weights $w$ the posterior probability of a patch assignment is approximated by:

$$Pr(h|x) \approx \frac{\exp(-\sum_i w_i^h (v_i^T x)^2 + const_h)}{\sum_k \exp(-\sum_i w_i^k (v_i^T x)^2 + const_k)} \tag{7}$$

We minimize the cross entropy between the approximate posterior probability and the exact posterior probability given by equation 1. The training is done on 500 mini-batches of 10K clean image patches each, taken randomly from the 200 images in the BSDS training set. We minimize the training loss for each mini-batch using 100 iterations of `minimize.m` [15] before moving to the next mini-batch.

Results of the training are shown in figure 3. Unlike the eigenvectors of the GMM covariance matrices which are often global Fourier patterns or edge filters, the learned vectors are more localized in space and resemble Gabor filters.

generatively trained:     discriminatively trained:

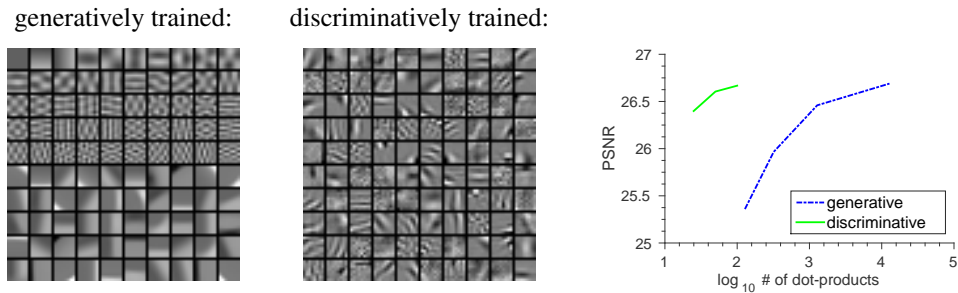

Figure 3: Left: A subset of the $200 \times 64$ eigenvectors used for the full posterior calculation. Center: The first layer of the discriminatively trained gating network which serves as a shared pool of 100 eigenvectors. Right: The number of dot-products versus the resulting PSNR for patch denoising using different models. Discrimintively training smaller gating networks is better than generatively training smaller GMMs (with less components).

Figure 1 compares the gating performed by the full network and the discriminatively trained one. Each pixel shows the predicted component for a patch centered around that pixel. Components are color coded so that dark pixels correspond to components with low variance and bright pixels to high variance. The colors denote the preferred orientation of the covariance. Although the gating network requires far less dot-products it gives similar (although not identical) gating.

Figure 4 shows sample patches arranged according to the gating with either the full model (top) or the gating network (bottom). We classify a set of patches by their assignment probabilities. For 60 of the 200 components we display 10 patches that are classified to that component. It can be seen that when the classification is done using the gating network or the full posterior, the results are visually similar.

The right side of figure 3 compares between two different ways to reduce computation time. The green curve shows gating networks with different sizes (containing 25 to 100 vectors) trained on top of the 200 component GMM. The blue curve shows GMMs with a different number of components (from 2 to 200). Each of the models is used to perform patch denoising (using MAP inference) with noise level of 25. It is clearly shown that in terms of the number of dot-products versus the resulting PSNR, discriminatively training a small gating network on top of a GMM with 200 components is much better than a pure generative training of smaller GMMs.

gating with the full model

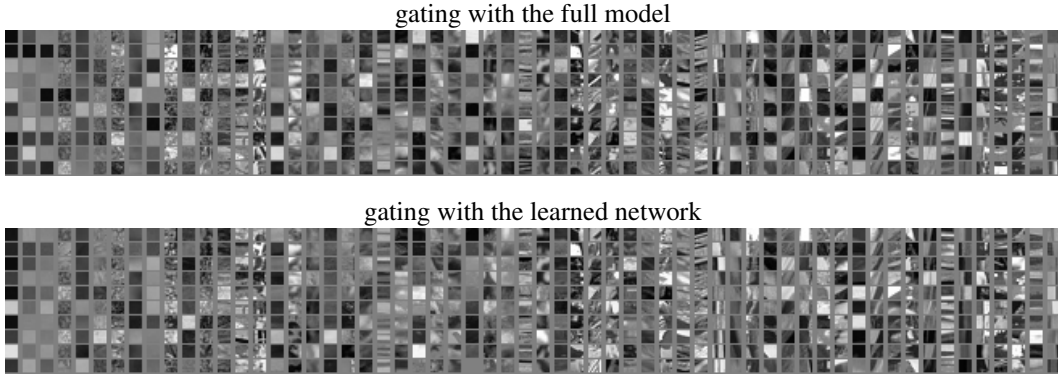

gating with the learned network

Figure 4: Gating with the full posterior computation vs. the learned gating network. Top: Patches from clean images arranged according to the component with maximum probability. Every column represents a different component (showing 60 out of 200). Bottom: Patches arranged according to the component with maximum gating score. Both gating methods have a very similar behavior.

## 4 Results

We compare the image restoration performance of our proposed method to several other methods proposed in the literature. The first class of methods used for denoising are "internal" methods that do not require any learning but are specific to image denoising. A prime example is BM3D. The second class of methods are generative models which are only trained on clean images. The original EPLL algorithm is in this class. Finally, the third class of models are discriminative which are trained "end-to-end". These typically have the best performance but need to be trained in advance for any image restoration problem.

In the right hand side of table 1 we show the denoising results of our implementation of EPLL with a GMM of 200 components. It can be seen that the difference between doing the full inference and using a learned gating network (with 100 vectors) is about 0.1dB to 0.3dB which is comparable to the difference between different published values of performance for a single algorithm. Even with the learned gating network the EPLL's performance is among the top performing methods for all noise levels. The fully discriminative MLP method is the best performing method for each noise level but it is trained explicitly and separately for each noise level.

The right hand side of table 1 also shows the run times of our Matlab implementation of EPLL on a standard CPU. Although the number of dot-products in the gating has been decreased by a factor of

| $\sigma$ | 20 | 25 | 30 | 50 | 75 |
|---|---|---|---|---|---|
| **internal** | | | | | |
| BM3D[22] | | 28.57 | | 25.63 | |
| BM3D[1] | | 28.35 | | 25.45 | 23.96 |
| BM3D[6] | 29.25 | | 27.32 | 25.09 | |
| LSSC[22] | | 28.70 | | 25.73 | |
| LSSC[6] | 29.40 | | 27.39 | 25.09 | |
| KSVD[22] | | 28.20 | | 25.15 | |
| **generative** | | | | | |
| FoE[22] | | 27.77 | | 23.29 | |
| KSVDG[22] | | 28.28 | | 25.18 | |
| EPLL[22] | | 28.71 | | 25.72 | |
| EPLL[1] | | 28.47 | | 25.50 | 24.16 |
| EPLL[6] | 29.38 | | 27.44 | 25.22 | |
| **discriminative** | | | | | |
| $CSF^5_{7\times7}$[18] | | 28.72 | | | |
| MLP[1] | | 28.75 | | 25.83 | 24.42 |
| FF[6] | 29.65 | | 27.48 | 25.25 | |

| EPLL with different gating methods | | | | |
|---|---|---|---|---|
| $\sigma$ | 25 | 50 | 75 | sec. |
| full | 28.52 | 25.53 | 24.02 | 91 |
| gating | 28.40 | 25.37 | 23.79 | 5.6 |
| $gating_3$ | 28.36 | 25.30 | 23.71 | 0.7 |

full: naive posterior computation.
gating: the learned gating network.
$gating_3$: the learned network calculated with a stride of 3.

Table 1: Average PSNR (dB) for image denoising. Left: Values for different denoising methods as reported by different papers. Right: Comparing different gating methods for our EPLL implementation, computed over 100 test images of BSDS. Using a fast gating method results in a PSNR difference comparable to the difference between different published values of the same algorithm.

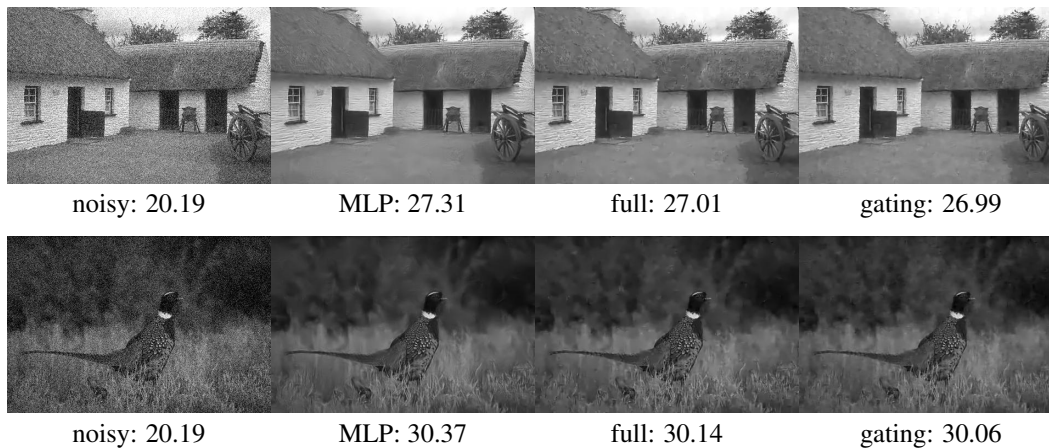

noisy: 20.19    MLP: 27.31    full: 27.01    gating: 26.99

noisy: 20.19    MLP: 30.37    full: 30.14    gating: 30.06

Figure 5: Image denoising examples. Using the fast gating network or the full inference computation, is visually indistinguishable.

128, the effect on the actual run times is more complex. Still, by only switching to the new gating network, we obtain a speedup factor of more than 15 on small images. We also show that further speedup can be achieved by simply working with less overlapping patches ("stride"). The results show that using a stride of 3 (i.e. working on every 9'th patch) leads to almost no loss in PSNR. Although the "stride" speedup can be achieved by any patch based method, it emphasizes another important trade-off between accuracy and running-time. In total, we see that a speedup factor of more than 100, lead to very similar results than the full inference. We expect even more dramatic speedups are possible with more optimized and parallel code.

Figures 5 gives a visual comparison of denoised images. As can be expected from the PSNR values, the results with full EPLL and the gating network EPLL are visually indistinguishable.

To highlight the *modularity* advantage of generative models, figure 6 shows results of image deblurring using the same prior. Even though all the training of the EPLL and the gating was done on clean sharp images, the prior can be combined with a likelihood for deblurring to obtain state-of-the-art deblurring results. Again, the full and the gating results are visually indistinguishable.

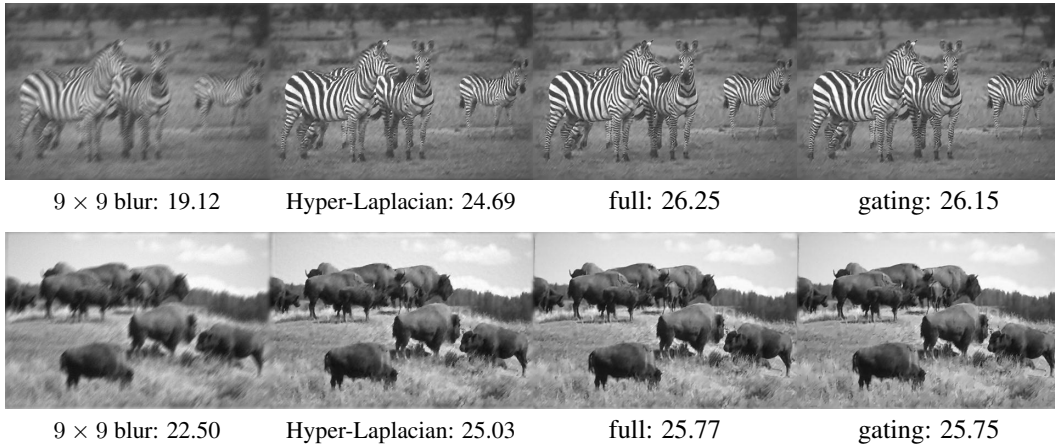

| $9 \times 9$ blur: 19.12 | Hyper-Laplacian: 24.69 | full: 26.25 | gating: 26.15 |

| $9 \times 9$ blur: 22.50 | Hyper-Laplacian: 25.03 | full: 25.77 | gating: 25.75 |

Figure 6: Image deblurring examples. Using the learned gating network maintains the modularity property, allowing it to be used for different restoration tasks. Once again, results are very similar to the full inference computation.

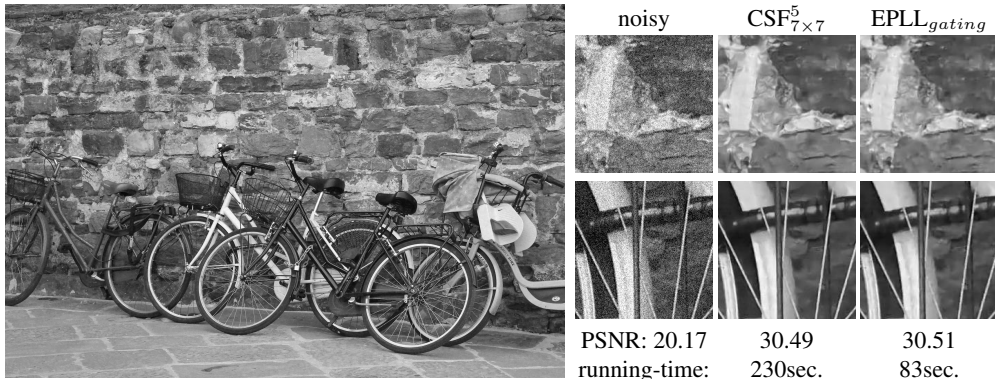

| | noisy | $\text{CSF}^5_{7\times7}$ | $\text{EPLL}_{gating}$ |
|---|---|---|---|
| PSNR: | 20.17 | 30.49 | 30.51 |
| running-time: | | 230sec. | 83sec. |

Figure 7: Denoising of a 18mega-pixel image. Using the learned gating network and a stride of 3, we get very fast inference with comparable results to discriminatively "end-to-end" trained models.

Finally, figure 7 shows the result of performing resotration on an 18 mega-pixel image. EPLL with a gating network achieves comparable results to a discriminatively trained method (CSF) [18] but is even more efficient while maintaining the modularity of the generative approach.

## 5   Discussion

Image restoration is a widely studied problem with immediate practical applications. In recent years, approaches based on machine learning have started to outperform handcrafted methods. This is true both for generative approaches and discriminative approaches. While discriminative approaches often give the best performance for a fixed computational budget, the generative approaches have the advantage of modularity. They are only trained on clean images and can be used to perform one of an infinite number of possible resotration tasks by using Bayes' rule. In this paper we have shown how to combine the best aspects of both approaches. We discriminatively train a feed-forward architecture to perform the most expensive part of inference using generative models. Our results indicate that we can still obtain state-of-the-art performance with two orders of magnitude improvement in run times while maintaining the modularity advantage of generative models.

**Acknowledgements**

Support by the ISF, Intel ICRI-CI and the Gatsby Foundation is greatfully acknowledged.

## Footnotes

[1]For non-local noise models like in image deblurring there is an additional factor of the square of the kernel dimension. If the kernel dimension is in the order of $d$, the mixing step performs $d$ dot-products or $d^2$ flops.

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
