[Supplementary Material]

# The Return of the Gating Network: Combining Generative Models and Discriminative Training in Natural Image Priors

# Supplementary Material

**Dan Rosenbaum**
School of Computer Science and Engineering
Hebrew University of Jerusalem
www.cs.huji.ac.il/~danrsm

**Yair Weiss**
School of Computer Science and Engineering
Hebrew University of Jerusalem
www.cs.huji.ac.il/~yweiss

## 1 Image restoration using EPLL with Gaussian mixtures

Modeling image patches with Gaussian mixtures has proven to be very effective for image restoration. The model is first trained by maximizing the likelihood over a training set of clean image patches. Then, it can be used as a prior model of clean patches in different image restoration tasks. In the EPLL framework [4], for every specific restoration task, one needs to define a noise model $Pr(y|x)$, and maximize the a-posteriori probability of clean patches given the noisy image. Using a simplifying assumption that all patches are indepedent yields the following cost to minimize:

$$J(x) = -\sum_i \log Pr(P_i x) - \lambda \log Pr(y|x) \qquad (1)$$

where $P_i$ is a matrix that extracts the i'th patch of the image $x$, and $\lambda$ is a parameter that compensates for the simplifying assumption. Nevertheless, minimizing this simplified cost is still hard because of the fact that overlapping patches must agree on their shared pixels. One way to solve this is by using a quadratic penalty method [1, 4, 2], which allows patches to disagree but gives an additional penalty to the disagreement:

$$J_{split}(x, \{z_i\}) = -\sum_i \log Pr(z_i) - \lambda \log Pr(y|x) + \frac{\beta}{2} \sum_i \|z_i - P_i x\|^2 \qquad (2)$$

This cost is minimized by alternatingly solving for every patch $z_i$ and the whole image $x$. By increasing the penalty parameter $\beta$ throughout the iterations, the splitting cost converges to the original EPLL cost.

When the prior is a Gaussian mixture, the probability of a patch is computed by $Pr(z_i) = \sum_h \pi_h \mathcal{N}(z_i; \mu_h, \Sigma_h)$, and finding the patch with maximum probability can be performed using the expectation-maximization (EM) algorithm [3]. This is again an iterative process that involves two steps. In the E-step, we compute the probability of the hidden variable $h$ (the component of the mixture) given a current estimate of the clean patch. In the M-step, the noisy patch is filtered assuming it comes from the component with maximum probability. The complete process for solving the EPLL with a GMM prior is described in algorithm 1.

## References

[1] Donald Geman and Chengda Yang. Nonlinear image recovery with half-quadratic regularization. *Image Processing, IEEE Transactions on*, 4(7):932–946, 1995.

---

**Algorithm 1** EPLL with a GMM prior

---

Input: a noisy image $y$ with $n$ pixles.

       a GMM prior with $K$ components over patches of $d$ pixels.

       each component $h$ has zero mean, a covariance $C_h$ and a mixing weight $\pi_h$.

set initial image estimate $x = y$

**repeat**

    extract patches $\{x_i\}$ from current image estimate $x$

    **repeat**

        for every patch set the initial estimate: $z_i = x_i$

$$h_i = \operatorname{argmax}_h \log \pi_h - \tfrac{1}{2} z_i^\top C_h^{-1} z_i \qquad\qquad \{\text{gating. } \#dotp = d \times K\}$$

$$z_i = C_h \left( C_h + \tfrac{1}{\beta} I \right)^{-1} x_i \qquad\qquad \{\text{filtering. } \#dotp = d\}$$

    **until** convergence

    form an image $z$ by averaging all overlapping patches $\{z_i\}$.

    $x = \operatorname{argmin}_x -\tfrac{\lambda}{d} \log Pr(y|x) + \tfrac{\beta}{2}(x^\top x - x^\top z) \qquad \{\text{mixing. } \#dotp = 1\}$

**until** convergence

($\#dotp$ refers to the number of $d$ dimensional dot-products per patch)

---

1st layer: shared "eigenvectors"      2nd layer: the "eignvalues"      estimated mixing weights

Figure 1: The two layers of the discriminatively trained gating network. The first layer serves as a shared pool of eigenvectos. The second layer is a weight matrix: every line represents the coefficients of all the eigenvectors for a given component. The estimated mixing weights show that the gating statistics of the trained model are similar to the original one.

[2] Dilip Krishnan and Rob Fergus. Fast image deconvolution using hyper-laplacian priors. In *NIPS*, volume 22, pages 1–9, 2009.

[3] Effi Levi. *Using natural image priors-maximizing or sampling?* PhD thesis, The Hebrew University of Jerusalem, 2009.

[4] Daniel Zoran and Yair Weiss. From learning models of natural image patches to whole image restoration. In *Computer Vision (ICCV), 2011 IEEE International Conference on*, pages 479–486. IEEE, 2011.