[Reviews · NeurIPS 2015]

Submitted by Assigned_Reviewer_1

The basic idea is this: a popular prior for modeling natural image patches is the GMM. A computationally expensive step in (usually iterative) inference algorithms when using such models is computing the posterior mixture assignments---i.e., determining which component a particular patch belongs to. This usually requires computing the relative likelihood of the patch belonging to each component, by the expensive process of computing the norm of the patch with respect to each component's covariance matrix.

Instead, the authors propose learning a smaller gating network that computes these assignments (strictly speaking, the posterior mixing probabilities) using a neural network with a single hidden layer, with a relatively small number of units. The weights of this network are learned to minimize the KL divergence with respect to the true posterior distribution for a training set of patches.

Essentially, the authors propose training a classifier to predict the component assignments, and show that this discriminatively-trained classifier can replicate the behavior of an explicitly computed posterior from a generative model, but with far fewer computations. Moreover, since this can be seen as a "drop-in" replacement for a computation step that deals only with the prior, it can be trained once and used for inference for different tasks with different likelihood terms. This second property gives the approach an advantage over "end-to-end" discriminative training, which requires re-training for each task (for example, even in the case of de-blurring, such methods would require to be retrained for every blur kernel).

Overall, the paper is based on a novel and interesting idea that is well executed and reasonably evaluated. Indeed, I expect that it has the potential to be used in applications beyond image restoration. I am inclined to recommend acceptance, but would like the authors to comment on the following issues in the rebuttal/final manuscript:

1. The description of the number of computations in terms of dot products is a little confusing. For instance, in the traditional GMM case, the paper says the first layer computes d * K dot-products, and the second computes K dot-products. In this case, the length of the vectors (for which the dot-product is computed) is equal for both layers.

But in the proposed gating network, it is not. The first layer computes N dot-products of length d, and in the second, it computes K dot-products of length N. When in lines 241-243, the paper says that "we set the number of dot-products to be 100", it is not clear whether this refers to the total number of dot-products, or those only in the first layer (I suspect that this is the latter---but in this case, note that "the second layer" now requires computing K dot-products over much larger vectors).

It would be a *lot* clearer if the discussion was in terms of the number of multiplies and adds, and the number of neurons in the hidden layer (i.e., the number of v_i's) was explicitly mentioned.

2. I would have liked to see an evaluation (of say just denoising) when using different numbers of mixture components. An interesting question is: to what extent does one need to increase the number of neurons in the hidden layer as the number of mixture components goes up ? Or equivalently, how does the proposed approach change the accuracy-time trade-off for choosing the number of mixture components ?
Summary: The paper introduces a computationally-efficient alternative to compute posterior mixture-component assignments during inference in image restoration, when using Gaussian mixture model (GMM) priors for image patches.

Submitted by Assigned_Reviewer_2

This paper combines a generative and a discriminative approach to achieve state-of-the-art image restoration system with two orders of magnitude faster computation.

The idea of combining an unsupervised generative front panel, and then to use a linear combinations of the activated "primitives" is very old (a citation of Hertz, Krogh & Palmer 1991 or some earlier reference would be nice), but the particular formulation of the problem is new. The paper uses straightforward components and standard denoising measures to validate the results. I would have liked to see some analyses that show the extent to which the

new gating network is inferior to the original one or some deeper analysis of what has changed due to the net gating.
Summary: A straightforward implementation of an old idea with respectable results.

Submitted by Assigned_Reviewer_3

How fast is denoising with the MLP? If it is faster, the only advantage of denoising with GMMs would be its modularity. But I imagine that in real-world applications, the modularity of generative models is rarely a decisive advantage since the noise model will be known or will be one of a handful of choices (and not specified by a user, for example). Should I be wrong about this, it might be nice to include an example in the paper where modularity is of practical importance.
Summary: The story of the paper is very clear (using discriminative training to speed up denoising with generative models, in this case a GMM) and it is for the most part very well written. The approach is simple but the results are convincing.

Submitted by Assigned_Reviewer_4

The paper proposes a fast approximate inference approach for MAP estimation with the generative EPLL method [21]. To that end, discriminative training is used to replace the computationally expensive update step of the latent variables during alternating optimization. The experimental results show that while being 1-2 orders of magnitude faster, this approximate inference procedure causes only a relatively modest drop in performance as compared to previously-used ("exact") EPLL inference.

# Positive I very much agree with the agenda of the paper to retain the benefits of generative approaches, as discussed at the beginning of the paper. As far as I see, the paper unfortunately doesn't really propose a generic method, since it only addresses a very specific problem of inference with the EPLL method. However, it is an interesting approach to use discriminative methods to approximate expensive steps during generative inference. Have you thought about how this can be used more generally? This would really interest me.

It is interesting that EPLL yields similar performance with the proposed approximative inference procedure compared to the "exact" one. This has value, since EPLL yields state-of-the-art results and can now also be applied to large images with the proposed inference method (Figure 7 is quite impressive); I think the paper should emphasize this more.

The puzzling part to me is why EPLL is seemingly the only generative approach that is competitive with discriminative methods; this would be interesting to understand.

# Negative I think that the paper over-promises its contributions. I was really excited in the beginning while reading the introduction, but was then somewhat disappointed at the end. This is because the paper is somewhat misleading due to introduction being very generic and promising a general method to combine generative and discriminative approaches. However, the paper doesn't propose a "deep" integration between generative and discriminative methods. It requires standard generative training of the image prior, then trains a discriminative predictor to approximate expensive steps of generative inference.

Furthermore, the paper then addresses only a very specific problem for performing MAP estimation with the EPLL method, which is as far as I see not easily applicable beyond this. Moreover, other generative approaches with filter-based image priors don't have the problem that updating the latent variables is computationally expensive [e.g., Krishnan and Fergus, "Fast image deconvolution using hyper-Laplacian priors", NIPS 2009].

Although not really important for the actual contribution of the paper, the discussion of discriminative and generative approaches starting in l. 111 is not very nuanced. There are other things to consider than efficiency of test-time inference and "modularity", such as generative approaches being able to use different loss functions at test time, or performing marginalization as a principled approach to handle unknown random variables.

# Clarity - The explanation of the gating network in sec. 3 is hard to understand, I think it needs some work to make it more clear. - Table 1 doesn't say for which test data these denoising results have been obtained. Are these different test sets in each paper? - Typo in Eq. 7: the weight w_i^j should be w_i^k in the denominator. - Typo in l. 102: "Hel-Or used..." -> "Hel-Or and Shaked used..." - Figure 2 is not clear since the different parts of the figure are not labeled. Also note that there seems to be a missing connection in the gating network on the right side: the bottom left node should be connected to the middle right node. - I find the title "return of gating network" not really appropriate. Also, "combining generative models and discriminative training in natural image priors" indicates to me that a generic method is proposed in the paper, which is not the case.

# Miscellaneous - Note that "modularity" can also be achieved with discriminative approaches by training with a likelihood component that can later be replaced, which has been done by [Chen et al., "Revisiting loss-specific training of filter-based MRFs for image restoration", GCPR 2013, doi:10.1007/978-3-642-40602-7_30]. They trained a CRF via loss-based training for image denoising and then used it without re-training for deconvolution and super-resolution by swapping the likelihood term. - Note that EPLL inference doesn't actually use half-quadratic splitting as introduced by Geman and colleagues (for filter-based MRFs), cf. [Geman and Reynolds, "Constrained restoration and the recovery of discontinuities", PAMI 1992; Geman and Yang, "Nonlinear image recovery with half-quadratic regularization", TIP 1995]. I think the used method is best characterized as a quadratic penalty method for approximate constrained optimization [cf. Nocedal and Wright, "Numerical Optimization", Sec. 17.1].
Summary: Although technically simple, the paper makes an important contribution since it yields a generative image restoration approach that is competitive with state-of-the-art discriminative methods, both in terms of restoration quality and speed of inference.

Author Feedback
Author rebuttal: We thank the reviewers for their helpful remarks and are happy they found our paper novel, interesting and well-executed.

In response to reviewer 1:
As you suggest, we will add in the final version a more detailed evaluation of the performance as a function of the number of components and number of filter vectors.
We will also make the dot-product analysis clearer.

In response to reviewer 3:
We agree that it will be better to show how our method can be used with more models beyond the GMM. We have already used it with the Karklin and Lewicki model ("Emergence of complex cell properties by learning to generalize in natural scenes" 2009), which is another successful model for natural image patches (although not as widely used as the GMM). Our results show that also with this model, by using a discriminatively trained network for the inference of hidden variables, we achieve comparable denoising results with a speedup of 10 to 100 times. We will add this result to the final version.

As you suggest, we will make the discriminative vs. generative discussion more nuanced. We will mention the ability of the generative model to work with different loss functions in test time. We will also rename the "half-quadratic splitting" method as a "quadratic penalty" method.
The Chen et. al paper you refer to is very interesting. It can also be viewed as a generative model that is trained with a procedure similar to noise-contrastive divergence rather than maximum likelihood. We will mention this in the final version.

In response to reviewer 5:
We do not agree that in real-world applications the noise model will only be one of a handful of choices. In fact, image degradation in the real world can occur as a result of many factors including camera shake, quantization, compression artifacts, physical noise and more. Training an end-to-end system for all these combinations is completely unrealistic.